# Thermal Ablation Damage Analysis of CFRP Suffering from Lightning Based on Principles of Tomography

**DOI:** 10.3390/ma13225159

**Published:** 2020-11-16

**Authors:** Bin Li, Fei Chang, Yao Xiao, Xiaolong Wei, Weifeng He, Yueke Ming

**Affiliations:** 1Aeronautics Engineering College, Air Force Engineering University, Xi’an 710038, China; lee_bean@126.com (B.L.); changefly@163.com (F.C.); hehe_coco@163.com (W.H.); 2Aviation Maintenance NCO Academy, Air Force Engineering University, Xinyang, 464031, China; 3State Key Laboratory for Manufacturing Systems Engineering, Xi’an Jiaotong University, Xi ’an 710054, China; mingyueke@foxmail.com

**Keywords:** carbon fiber reinforced polymer (CFRP), thermal ablation, lightning strike, finite element analysis (FEA), computerized tomography (CT)

## Abstract

Coupled electrical–thermal finite element analysis (FEA) models are widely adopted to analyze the thermal ablation damage of carbon fiber reinforced polymer (CFRP) caused by lightning, but it is still difficult to analyze the ablation due to its complex space geometry. According to the principle of computerized tomography (CT), tomographic images of FEA models’ temperature fields with different thicknesses were obtained to calculate the mass loss and compare the damage morphology. The four areas including Area 0, Area I, Area II, and Area III; were separated from the temperature fields in terms of different vaporization and pyrolysis temperature ranges of carbon fiber (CF) and resin matrix. Ablation mass losses were calculated by pixel statistics and tomographic intervals, which were consistent with the experimental results. The maximum ablation area of unprotected CFRP was found on the tomography images of 50 μm rather than the surface by comparing tomographic images with different thickness due to the influence of the thermal radiation, but this effect was not found in CFRP protected by copper mesh. Some other phenomena, including continuous evolutions of ablation areas and the influence of the intersection angle on the direction of the ablation extension, were also discovered.

## 1. Introduction

The application research of carbon fiber reinforced polymer (CFRP) is in a stage of rapid development, and its application scope has achieved a leap from the secondary and small bearing component of aircraft to the main and large bearing component, and even the emergence of full composite aircraft [1,2,3]. Compared with metal materials, CFRP has better properties in terms of specific strength, specific modulus, corrosion resistance, fatigue resistance, and thermal stability. It is normal for an aircraft to be struck by lightning in flight, with the generally accepted probability of once every 3000 h. The traditional metal structures of the entire aircraft can overlap each other to form a good conductor, which can smoothly guide the lightning current, generate the "Faraday cage" effect, and protect the airborne equipment [4,5]. However, with the increasing application of the resin matrix composites in aircraft, its conductivity is gradually decreased, which makes these aircraft more vulnerable to lightning than before.

Carbon fiber and resin are the main components of CFRP, while some impurities can appear in CFRP such as curing agent and defoaming agent, etc. In fact, the amount of impurities is very small, and their impact on the overall properties of CFRP is negligible [3]. It is believed that the lightning damage of CFRP is mainly caused by the intense Joule effect of lightning current. A large amount of heat is generated in this process due to the intense electric heating of CFRP. Ablation and many other indirect effects are caused, including the recoil effect of resin vaporization, delamination damage caused by thermal expansion of CFRP [6,7]. Of course, the overpressure, electromagnetic force, and high temperature of the lightning current channel itself also play a secondary role [8]. Gammon earlier described the lightning damage of CFRP on the aircraft and observed it through a microscopic image [9]. Ferraboli utilized a lightning current exciter to discharge on CFRP in the laboratory earlier. The damage was observed under the microscope by ultraviolet fluorescence staining, and the change in residual strength was measured [10]. With the development of high-speed camera technology, the observation method of lightning damage process on CFRP has also gradually improved. Ben Wang and Qianshan Xia took advantage of high-speed cameras to acquire the process of adhesion to CFRP after a lightning current excitation and subsequent resin matrix cooling after ablation [11,12]. Hirano and Shintaro channeled a high-speed camera into photographing the shock wave generated by the electrode discharge, and the damage of CFRP incurred by the shock wave was studied through experiments [13,14]. Of course, measuring the size of ablation area, weighing the mass loss, and testing the residual strength of specimens after lightning strikes are still important and conventional methods to determine the lightning damage [15,16]. Copper mesh is one of the most important lightning protection methods for CFRP, and more research and applications have been made [17,18,19]. However, in the finite element analysis and a large number of experiments, it is still necessary to have accurate methods of calculating the mass loss and mechanical strength damage of the model. 

Computerized tomography (CT) is a technique that uses the radiation emitted by the detector to scan the section of the object one by one in order to form continuous tomographic images and observe its internal structure. This technique has been widely applied in medicine. Many scholars made use of it to calculate the volume of thyroid and tumor [20,21], and the 3D reconstruction based on CT has also been combined with TEM to analyze the microscopic details in bone research [22]. Yoshiyasu Hirano used the micro X-ray CT scanner to measure the damage depth of CFRP [23]. Katunin used X-ray to obtain CT images of CFRP laminates on different sections [24]. Yugang Duan studied the morphology of delamination damage through CT images [15]. 

In order to more accurately analyze the quality loss and the morphology of ablation with complex geometry, the fundamental idea of CT, which is to cut a continuous 3D object into discrete 2D plane graphics, was applied in the finite element analysis (FEA) models processing.

## 2. FEA Models and Experiments

### 2.1. Specimens and Their FEA Models

According to the standard ASTM D7137 and principle of lightning ablation [25,26,27], two types of specimens were designed and fabricated, including unprotected CFRP laminates (Group A) and copper-mesh-protected CFRP laminates (Group B). The ply sequence of both types of laminates is [45/-45/0/90/90/-45/0/45/0/90/-45/45] with the single ply thickness of 0.15 mm and the general size of 150 mm × 100 mm × 3.6 mm. The copper mesh is vertically woven and its mesh number is 1100 with the copper wire diameter of 0.25 mm. The appearance of the specimens is shown in Figure 1.

Corresponding to two groups of specimens, coupled electrical–thermal FEA models were established by using ABAQUS. Because of the lightning attached area, the central mesh of Group A model was locally designed to be denser to ensure the calculation accuracy. The model of Group B was composed of two parts including copper mesh and internal CFRP laminates of the same FEA model as Group A. The circular wire section in the model of copper mesh was converted into squares to generate hexahedral elements. All the above element types were DC3D8E (8-node linear, coupled electrical–thermal, and solid element). The appearance of FEA models is shown in Figure 2.

The boundary conditions and contact properties of FEA models were set according to the actual experimental conditions: The boundary heat flux of models was set to 0 W/m2, the surface thermal emissivity of CFRP laminates was 0.9 and that of the copper mesh was 0.78, the environmental temperature and the initial temperature of the specimens were both 25 °C, and the bottom and side potential in the models were 0V. There was often contact electrical and thermal resistance between different media. The electrical and thermal conductivity of the contact interface between copper mesh and CFRP laminates were reduced to 80% of the copper mesh at the same temperature [28]. The condition settings of the FEA models is shown in Figure 3.

### 2.2. Material Properties

The conductivity of CFRP is mainly determined by the carbon fiber (CF) inside, and the anisotropy of conductivity is determined by the different orientations of plies, but copper is isotropic material. The conductivity of CFRP and copper shows different trends with increasing temperature, which is because that the resin matrix in CFRP is an insulator which will melt when heated, while the copper is a conductor whose conductivity will decrease with increasing temperature. Property variations with temperature of CFRP and copper as shown in Table 1 and Table 2.

### 2.3. The Waveform of Lightning Currents

On the basis of SAE ARP-5412B, the lightning current waveforms were classified into 4 types: Component A (the first return stock), Component B (the intermediate current), Component C (the continuing current), and Component D (the subsequent return stroke). Typical lightning current waveforms are shown in Figure 4 [30].

Component D was selected as the lightning current load to analyze the direct impacts of CFRP lightning strikes. Six combinations of currents and protection modes are shown in Table 3.

### 2.4. Observation of the Ablation Damage

Ablation damages were first observed by temperature field of FEA models and CT, which are shown in Figure 5. The 3D reconstruction image of A20 was chosen, as other reconstructions were not sharp enough because of the warped layer and the complex structure of copper mesh both disrupting the propagation of the rays.

The FEA temperature field of A20 showed clearly the inner complex shape of ablation rather than a taper, reflecting that a more precise observing method was needed. The CF inside the CFRP could be clearly found by 3D reconstruction based on CT, but only the first ply was reconstructed, as much more continuous CT images will be required for other unnecessary inner plies without damage. In the single CT image of Figure 5b, the absence of copper mesh with high light was found, but the CFRP was still intact, which was consistent with the temperature field. Therefore, combining the advantages of CT and FEA, a new analysis method of ablative damage observation will be described in the following.

## 3. Processing of Tomography Images

### 3.1. Temperature Fields Zoning

Both CF and resin are organic polymer materials and belong to amorphous, whose vaporization and pyrolysis temperatures are not fixed. It is generally believed that 300 °C is the initial vaporize and pyrolysis temperature of the resin, and 573.5 °C is the completion temperature. The vaporize and pyrolysis of CF occur at 3042–3316 °C [31]. Four areas including Area Ⅰ, Area Ⅱ, and Area Ⅲ were divided from the temperature fields calculated from the coupled electrical–thermal FEA models, and the resin content in CFRP is 40%. Properties of different temperature areas are shown in Table 4.

The appearance of Area 0, Area I, Area II, and Area III also look different, 45° unidirectional CFRP in different temperature areas are shown in Figure 6.

Damage was not observed in Area 0, which still showed an intact appearance. Slight thermal ablation started to be established in Area I due to the fiber laying direction, which showed a part of resin was lost. Warpage and exposure of CF were found in Area Ⅱ due to the full vaporization and pyrolysis of resin; the mechanical properties in this area had almost been lost, although the CF seemed to be intact. Breakages and losses of CF were found in Area III, which indicated that the temperature here had reached the vaporization pyrolysis point of CF causing the most serious damage.

### 3.2. Calculation of Ablative Mass Based on Tomographic Images

The lightning ablation region of CFRP extends along different directions in different plies due to the anisotropy of CFRP, which leads to the complex geometric space of ablation. Ultrasonic scanning is usually channeled into finding the internal damage, while some other researchers have simplified it into an inverted cone to calculate the damage volume [32]. The ablative space of CFRP is segmented along the thickness direction, as shown in Figure 7.

Tomographic images at different thickness were extracted to discretize the continuous ablation space. The total ablative mass loss is expressed by Equation (1).
(1)$$m=\sum_{i=0}^{n}S_{i}^{Ι}\cdot h_{i}\cdot \rho \cdot 0.30+S_{i}^{\mathrm{II}}\cdot h_{i}\cdot \rho \cdot 0.45+S_{i}^{\mathrm{III}}\cdot h_{i}\cdot \rho \cdot 0.95$$
where *m* is the total ablative damage mass; $S_{i}^{I}$,$S_{i}^{\mathrm{II}}$, and $S_{i}^{\mathrm{III}}$ are the areas of Area I, II, and III in the image of Section *i*; *ρ* is the density of CFRP; *h_i_* is the distance between adjacent sections. 

If *h_i_*s are all equal between different adjacent sections, Equation (1) can be simplified as:(2)$$m=h_{i}\cdot \rho \cdot \sum_{i=0}^{n}\left(S_{i}^{I}\cdot 0.30\cdot S_{i}^{\mathrm{II}}\cdot 0.45+S_{i}^{\mathrm{III}}\cdot 0.95\right)$$

Pixel statistics were used to calculate the area of different areas in temperature fields of FEA. The relationship between the number of the pixels and the area is expressed as:(3)$$S_{i}^{I,\mathrm{II},\mathrm{III}}=S_{i}^{t}\cdot \frac{n_{i}^{I,\mathrm{II},\mathrm{III}}}{n_{i}^{t}}$$
where $S_{i}^{I,\mathrm{II},\mathrm{III}}$ and $S_{i}^{t}$ are the areas of Area I, II, or III and the total area of Section *I*; $n_{i}^{I,\mathrm{II},\mathrm{III}}$ and $n_{i}^{t}$ are the pixels’ number of Area I, II, or III and the total area of Section *i*.

When Equation (3) is introduced into Equation (2), the ablation mass loss is expressed as:(4)$$m=h\cdot \rho \cdot \frac{\sum_{i=0}^{n}S_{i}^{t}}{\sum_{i=0}^{n}n_{i}^{t}}\cdot \left(0.3\cdot \sum_{i=0}^{n}n_{i}^{I}\cdot 0.45\cdot \sum_{i=0}^{n}n_{i}^{\mathrm{II}}+0.95\cdot \sum_{i=0}^{n}n_{i}^{\mathrm{III}}\right)$$

## 4. Results

### 4.1. The Ablation Mass Loss

The distance between adjacent sections was set as 50 μm in the calculation of temperature fields of Group A, which represents that three tomographic images will be obtained in every ply. Each tomographic image was numbered as “*a*–*b*” where *a* is the ply number between 1 and 24 and *b* is the sections’ number in every ply between 1 and 3. The summary of tomographic images of Group A is shown in Figure 8.

The distance between adjacent sections was set as 5μm in the calculation of temperature fields of Group B due to the short damage depth of Group B and its significant variety along the thickness. A summary of the tomographic images of Group B is shown in Figure 9.

Images of different areas were selected through color recognition based on Figure 8 and Figure 9. The number of pixels was counted out and the area, volume, and mass of different areas were calculated according to Equations (2)–(4). The tomographic summary in different areas of A20 are shown in Figure 10.

Results of pixel statistics and ablation mass losses obtained by Equation (2)–(4) are shown in Table 5.

Mass losses of specimens were also obtained by weighing as shown in Table 6.

Variations of the mass losses of the six specimens and their FEA models demonstrated the same trends with the variations of currents, which will increase with the increase in the peak current. In Group A, the FEA results were less than the experimental results, because there were still any other forms of energy dissipation including air exchange, thermal convection, etc., in addition to thermal radiation during experiments. In Group B, the FEA results were significantly less than the experimental results due to copper mesh falling off at a high temperature [33].

### 4.2. The Ablation Morphology

The maximum ablation area of CFRP in Group A was found on the tomographic images of 50 μm instead of the surface by comparing tomographic images in different thickness. This phenomenon was called “subsurface effect”, whose main reason is the thermal radiation on surface; although the discharge time was just a few hundred microseconds, energy dissipation caused by the thermal radiation is considerable at a temperature difference of several thousand degrees Celsius. On the contrary, no thermal radiation to the external environment exists inside the CFRP, which counteracts the energy reduction caused by the weakening of the electric field, leading to the ablation area on the subsurface larger than on the surface. Surface temperature field images superimposed by tomographic images of 50 μm were found closer to the experimental results. The results of Group A are shown in Figure 11.

Unlike Group A, the “subsurface effect” of Group B was not found in Figure 9, which means that the maximum ablation area of CFRP still appears on the surface by comparing the tomographic images. Furthermore, the ablation area extension in all directions of Group B was almost the same, while that of Group A was not. The reason why these two led to different results is that the conductivity of copper is much higher than that of CFRP, so the copper mesh conducted most of the currents during lightning. The isotropic conductivity of the copper mesh results in a nearly circular ablation area in Group B. The Joule effect is concentrated on the copper mesh and its contact interface, leading to almost no current passing through underground surface of CFRP. The results of Group B are shown in Figure 12.

The ablation extension direction of copper-mesh-protected specimens is also influenced by the cross angle of copper wire. The conductivity of copper mesh will be anisotropic when the cross angle is not 90°, which will lead to different extension sizes of the ablation area in different directions. The ablation extension of rhombic braided copper mesh is exhibited in Figure 13 [30].

The FEA result of B20 was in good agreement with its experimental result, followed by B40 and B60, with all ablation occurring on the copper mesh and the CFRP surface formed from the intact CF of 45° in ultrasonic C-scan images. The maximum temperature of B20 was 343.6 °C, which was much lower than the vaporization temperature of copper but higher than its oxidation temperature, being consistent with the specimen of B20 only showing oxidation discoloration [34]. The ablation area and shape of B40 seemed the same in FEA and the experiments, while those of B60 were slightly different between the experiments and FEA. It can be inferred that although the maximum temperature of CFRP has not reached 3316 °C, the steam generated by the vaporization and pyrolysis of resin will cause the copper mesh to fall off at high temperatures and reduce its mechanical properties. 

Some other differences including the ablative pits of B20 and the oxidation discoloration and warping at the edge of B40 or B60 were also observed. The reason for the above phenomenon is that the actual current channel is a local energy concentration at the intersection of copper wires rather than a uniform plasma flow. The contact resistance between lapped copper wires or between specimens and fixtures also plays an important role, which will increase in ablation area of B40 and B60. The microscopic image of the central region of B20 specimen is shown in Figure 14.

### 4.3. Evolutions of Ablation Areas

In the past, the entire ply was usually scanned by ultrasound or sectioned by tomography images of FEA regardless of the ply thickness. Due to the limitation of the technical accuracy, discontinuity or mutation occurred in the evolution of ablation area, which was shown in Figure 15 [6,35].

Different from the above results, it was found that even in the same plies, the evolutions of ablation areas are continuous. The continuity of the evolution of ablation areas also conforms to the basic principle of "the macroscopic material change is continuous" in classical physics [36]. It was proved that the conductivity of different plies will affect its own electric field as well as the electric field of other plies, leading to the continuous change in the ablation area. Superimposed ablation area boundaries are shown in Figure 16.

Boundaries 1–1, 1–2, and 1–3 extended along the *l*_1_, becoming wider and shorter as the tomography deepens. The extended scale of Boundary 2–1 was located between Ply 1 and Ply 2, which looked like a square and symmetrical along *l*_2_, because it is located at the junction of the two plies. Boundary 2–2 and Boundary 2–3 were both symmetrical along *l*_3_, which become longer and narrower with the tomography deepening.

### 4.4. Fitting Analysis between Ablation Mass Losses and Currents

The integration of action and the charge transfer are the two main parameters of the currents, and the former can directly represent the current energy [37]. With the currents of same waveforms, the integration of action is proportional to the square of the amplitude, and the charge transfer is proportional to the amplitude. The quadratic curve fitting between ablation mass losses and peak currents can reflect the influence of the current action integral on the ablation damage. The quadratic curve fitting of mass losses between these two groups in experiments and FEA is shown in Figure 17.

The curve of “Δ*m* = A·*I*_peak_^2^” is used for fitting in Figure 17. The parameter analysis is demonstrated in Table 7.

The Coefficient A of the experiment was slightly less than that of the FEA by 14.75% in Group A, which indicates that the experimental results are in good agreement with the simulation results, while other forms of energy dissipation should be further considered in the FEA. However, the Coefficient A of the experiment is significantly greater than that of the FEA in Group B due to the surge of B60 results of the experiment, and the factors of copper mesh fracture and shedding should be further taken into consideration in the future.

## 5. Conclusions

(1)According to different degrees of vaporization and pyrolysis of carbon fiber and resin matrix at different temperatures, the temperature field of CFRP in lightning is divided into Area 0, Area I, Area Ⅱ, and Area Ⅲ with different density loss rates. The calculation accuracy will be improved if the temperature range is divided more finely without considering the simplicity of the calculation process.(2)The areas of Area I, Area II, and Area III in different tomographic images are calculated by pixel statistics. The total mass loss can be obtained by multiplying the area with the corresponding spacing, density, and mass loss rate. Ablation mass losses are well-matched with that of FEA. The calculation accuracy will increase by shortening the spacing without considering the simplicity of the calculation process.(3)The maximum ablative area of unprotected CFRP laminates appears underground rather than on the surface due to the surface thermal radiation, while this effect does not appear in copper mesh protected CFRP laminates, because the copper mesh bears most of the lightning current, leading to the ablation area being close to circular at the same time.(4)The ablative areas of CFRP gradually evolve with the increase in the thickness even in the case of the same ply, and the copper mesh can significantly reduce the anisotropy of ablation areas of CFRP. The anisotropy of ablation areas is closely associated with the cross angle of copper wire, where the ablation areas are close to circular if the angle is 90°, and the ablation areas are close to ellipse if the angle is not 90°.(5)The quadratic curves are used to fit ablation mass losses. Since the integral of current action is proportional to the lightning energy, the quadratic coefficient of the fitted curve can reflect the ablation energy.

## Figures and Tables

**Figure 1 materials-13-05159-f001:**
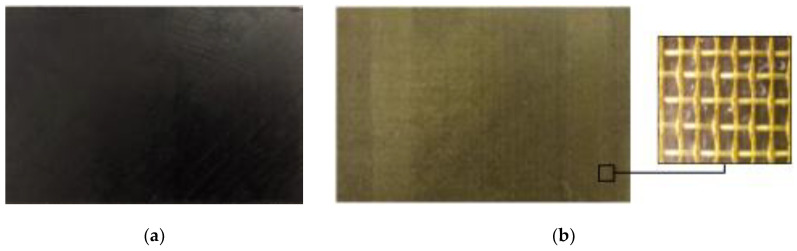
Appearance of specimens. (**a**) The unprotected carbon fiber reinforced polymer (CFRP) laminates (Group A); (**b**) the copper mesh protected CFRP laminates (Group B).

**Figure 2 materials-13-05159-f002:**
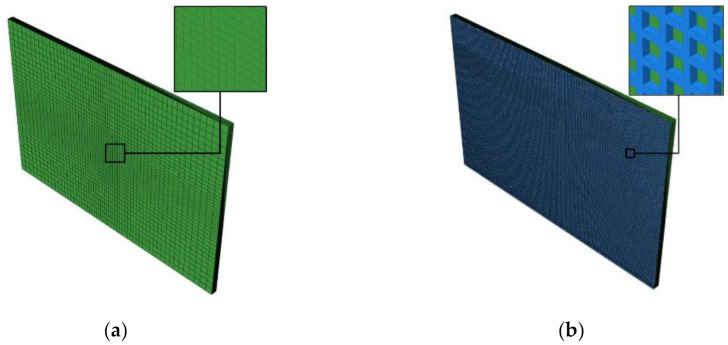
Appearance of finite element analysis (FEA) models. (**a**) The unprotected CFRP laminates (Group A); (**b**) The copper mesh protected CFRP laminates (Group B).

**Figure 3 materials-13-05159-f003:**
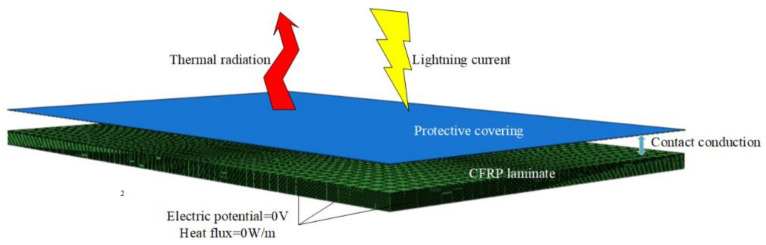
Conditions settings of the FEA models.

**Figure 4 materials-13-05159-f004:**
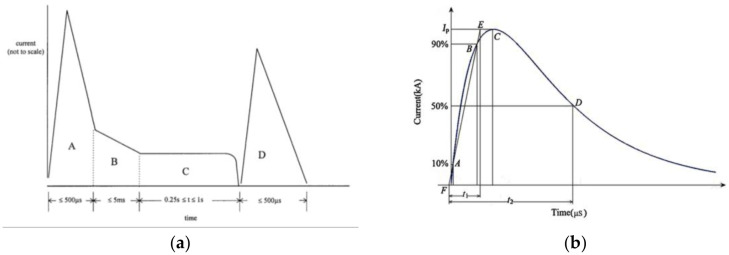
Typical lightning current waveforms: (**a**) 4 types of lightning current waveforms; (**b**) meanings of different symbols in Table 2.

**Figure 5 materials-13-05159-f005:**
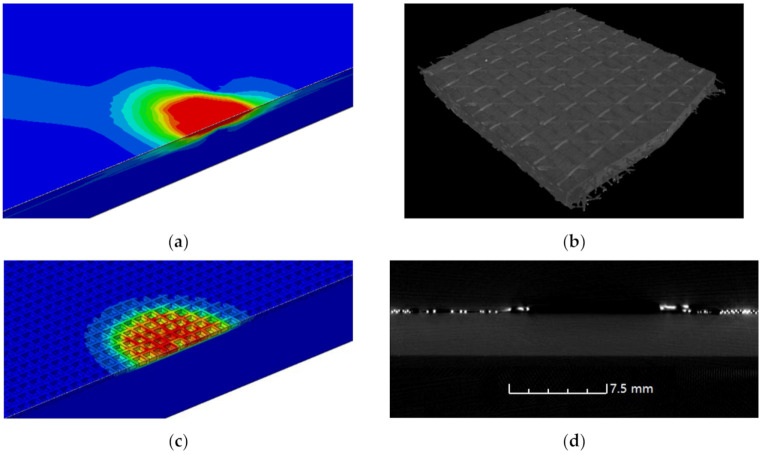
Ablation damage images of FEA models and computerized tomography (CT): (**a**) the temperature field of FEA model of A20; (**b**) 3D reconstruction based on CT of A20; (**c**) the temperature field of FEA model of B60; (**d**) the industrial CT image of B60.

**Figure 6 materials-13-05159-f006:**
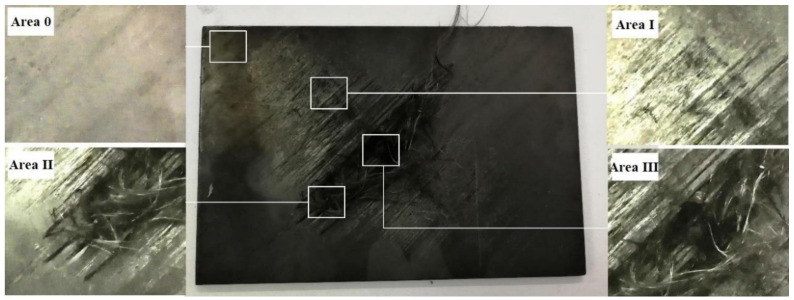
Different temperature areas of 45° unidirectional CFRP.

**Figure 7 materials-13-05159-f007:**
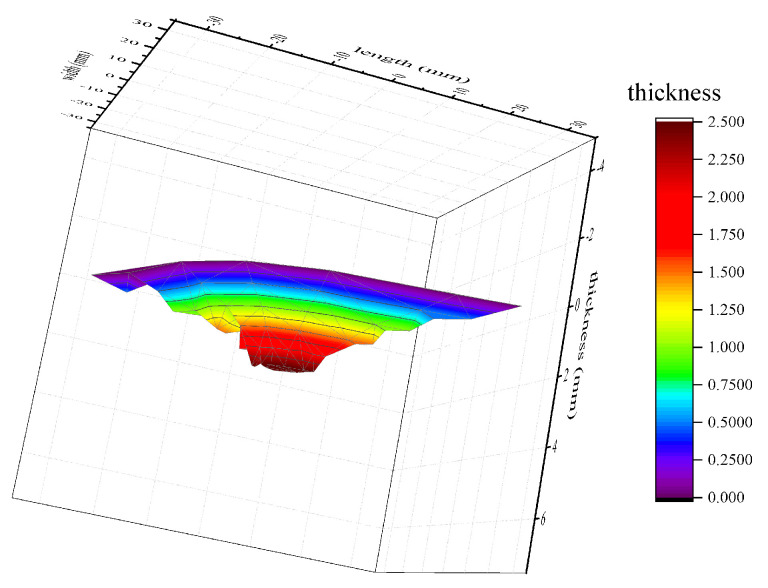
The ablative space of CFRP.

**Figure 8 materials-13-05159-f008:**
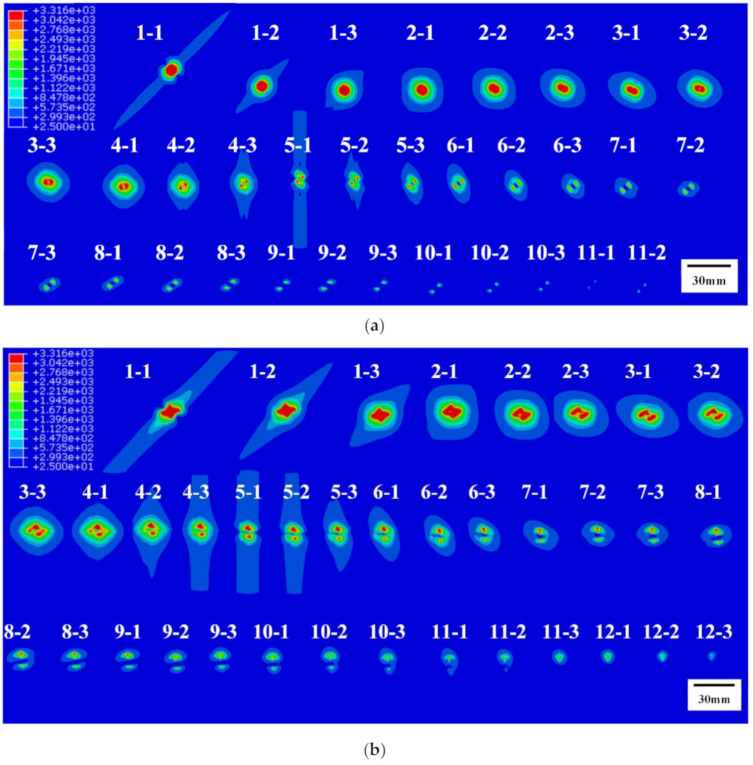
Tomography images’ summaries of different FEA models in Group A: (**a**) the A20s; (**b**) the A40s; (**c**) the A60s.

**Figure 9 materials-13-05159-f009:**
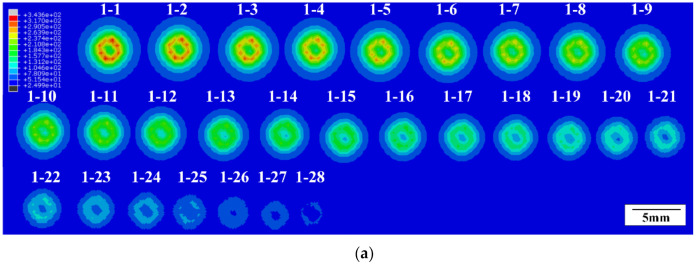
Tomography images’ summaries of different FEA models in Group B: (**a**) the B20s; (**b**) the B40s; (**c**) the B60s.

**Figure 10 materials-13-05159-f010:**
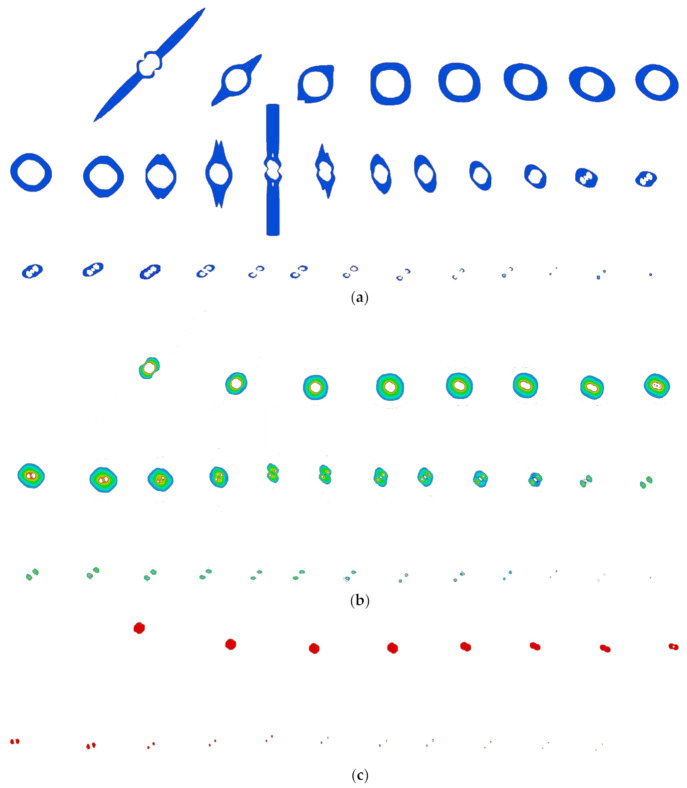
Different areas of tomography images’ summaries of A20: (**a**) Area I; (**b**) Area II; (**c**) Area III.

**Figure 11 materials-13-05159-f011:**
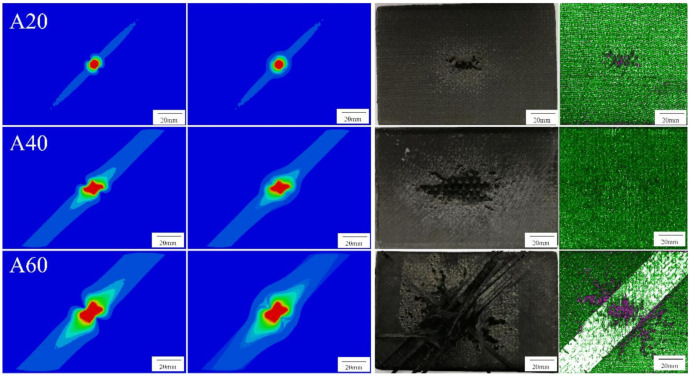
Results of Group A. Surface temperature field images of FEA, surface–subsurface superimposed temperature field images of FEA, specimen photos, and their ultrasonic C-scan images are shown from left to right.

**Figure 12 materials-13-05159-f012:**
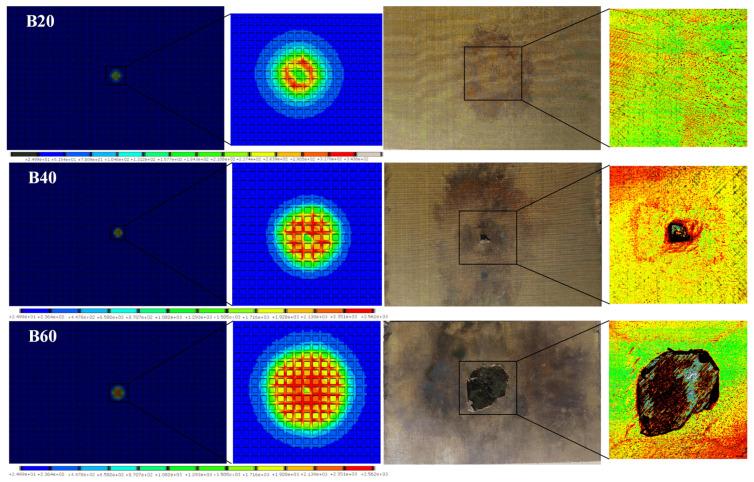
Results of Group B. Surface temperature field images of FEA, surface–subsurface superimposed temperature field images of FEA, specimen photos, and their ultrasonic C-scan images are shown from left to right.

**Figure 13 materials-13-05159-f013:**
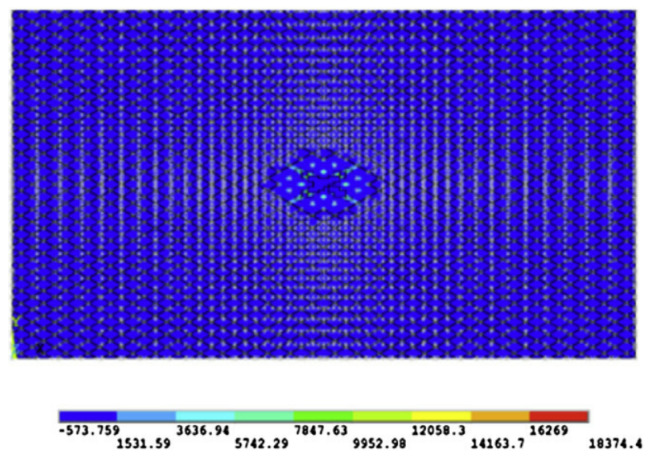
The ablation extension of rhombic braided copper mesh.

**Figure 14 materials-13-05159-f014:**
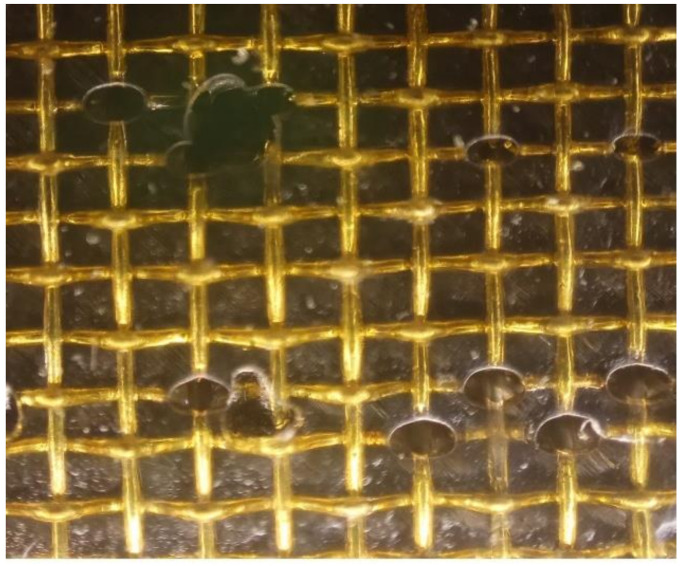
Microscopic image of central region of B20.

**Figure 15 materials-13-05159-f015:**
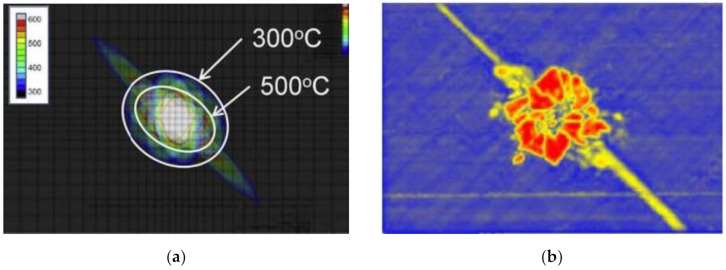
Results of FEA and experiments based on the whole ply analysis: (**a**) the result of FEA; (**b**) the result of the experiment.

**Figure 16 materials-13-05159-f016:**
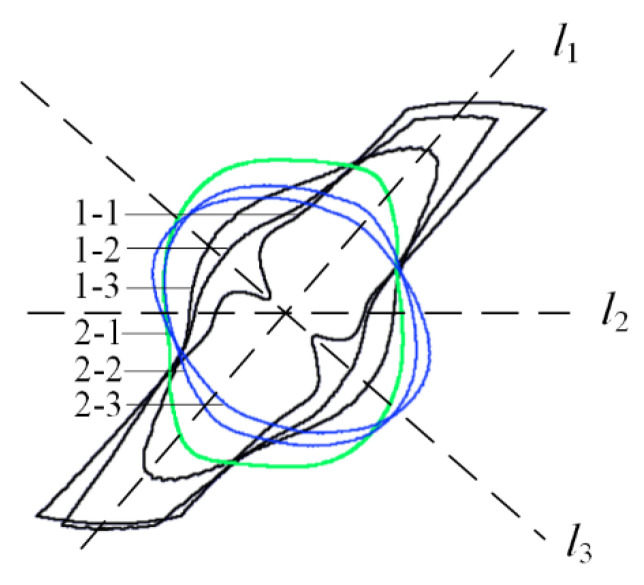
Ablation area boundaries of different tomographic images of the 1st and 2nd plies of A60. The angle of *l*_1_, *l*_2_, and *l*_3_ are 45°, 0°, and −45°.

**Figure 17 materials-13-05159-f017:**
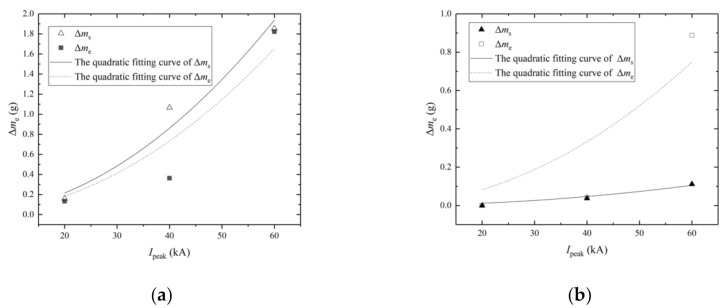
The quadratic curve fitting between ablation mass losses and peak currents in the experiments and FEA: (**a**) the result of Group A; (**b**) the result of Group B. Δ*m*_s_ and Δ*m*_e_ are ablation mass losses in FEA and the experiments, respectively.

**Table 1 materials-13-05159-t001:** Property variations with temperature of CFRP [29]. *T* is the temperature, *ρ* is the density, *C*_p_ is the specific heat, *σ*_x_, *σ*_y_, and *σ*_z_ are, respectively, the longitudinal, the transverse, and the in-depth electrical conductivity; *λ*_x_, *λ*_y_, and *λ*_z_ are, respectively, the longitudinal, the transverse, and the in-depth thermal conductivity.

*T* (°C)	*ρ* (J·kg^−1^·K^−1^)	*C*_p_ (J·kg^−1^·K^−1^)	*σ* (S·m^−1^)			*λ* (W·m^−1^·K^−1^)		
			*σ* _x_	*σ* _y_	*σ* _z_	*λ* _x_	*λ* _y_	*λ* _z_
25	1520	1065	29,300	77.8	0.07	11.8	0.609	0.609
300	1520	2100	29,300	77.8	0.07	11.8	0.609	0.609
400	1520	2100	29,300	778	7.94	2.608	0.18	0.18
600	1100	2100	29,300	2000	2000	1.736	0.1	0.1
3316	1100	2509	29,300	2000	2000	1.376	0.1	0.1
>3316	1100	5875	200	200	1e8	1.015	0.1	0.1

**Table 2 materials-13-05159-t002:** Property variations with temperature of copper [30]. *T* is the temperature, *ρ* is the density, *C*_p_ is the specific heat, *σ* is the electrical conductivity, *λ* is the thermal conductivity.

***T*** **(°C)**	***ρ*** **(J·kg^−1^·K^−1^)**	***C*** **_p_** **(J·kg^−1^·K^−1^)**	***σ*** **(S·m^−1^)**	***λ*** **(W·m^−1^·K^−1^)**
25	8950	385	58,140,000	401
500	8500	431	20,120,000	370
510	8490	431	4,651,000	339
1000	7945	490.952	3,704,000	150
2562	7600	490.952	2,227,000	180
>2562	7600	550	10^8^	180

**Table 3 materials-13-05159-t003:** Combinations of currents and protection modes. Meanings of *I*_p_, *t*_1_, and *t*_2_ are, respectively, shown in Figure 4b; *E* is the integral action; *Q* is the charge transfer.

Number	Protection Modes	Waveform	*I*_p_(kA)	*t*_1_ (μs)	*t*_2_ (μs)	*E* (A^2^s)	*Q* (C)
A20	Unprotected	D	20kA	17	168	46,316	4.13
A40			40kA			185,267	8.26
A60			60kA			416,851	12.38
B20	Copper mesh protected		20kA			46,316	4.13
B40			40kA			185,267	8.26
B60			60kA			416,851	12.38

**Table 4 materials-13-05159-t004:** Properties of different temperature areas in CFRP.

Area	Temperature Range	Vaporize and Pyrolysis	Density Loss (g/cm^3^)	Rate
0	<300 °C	None	0	0
I	300–573.5 °C	Part of resin	0.456	30%
II	573.5–3000 °C	All resin and little CF	0.684	45%
III	3042–3316 °C	All resin and most CF	1.444	95%

**Table 5 materials-13-05159-t005:** Results of pixel statistics and ablation mass losses. *S*^t^ is the total area; *n*^t^ is the total number of pixels; *n*^I^, *n*^II^, and *n*^III^ are, respectively, the total pixels’ number of Area I, Area II, and Area III; Δ*m*_CFRP_ and Δ*m*_Cu_ are, respectively, the ablation mass losses of CFRP and copper mesh.

Number	*S*^t^ (cm^2^)	*n* ^t^	*n* ^Ⅰ^	*n* ^Ⅱ^	*n* ^Ⅲ^	Δ*m*_CFRP_ (g)	Δ*m*_Cu_ (g)
A20	424.2258341	292,560	22,886	14,329	1209	0.159381	N/A
A40	1555.492525	101,898	13,771	9823	668	1.065747	N/A
A60	1783.498801	280,952	58,147	40,706	2787	1.853072	N/A
B20	28.89050453	380,554	142	0	0	0	0.000002
B40	12.81793357	356,420	76,116	80,918	0	0.001619	0.035803
B60	37.18111485	175,168	34,028	38,480	0	0.00444	0.106569

**Table 6 materials-13-05159-t006:** Mass losses of specimens. *m*_1_ is the original mass; *m*_2_ is the post-experiment mass; Δ*m* is the mass variations; Δ*m*’ is the differences compared with simulations.

Number	*m*_1_ (g)	*m*_2_ (g)	Δ*m* (g)	Δ*m*’ (g)
A20	90.9002	90.7664	0.1338	−0.025581
A40	98.5323	98.1691	0.3632	−0.702547
A60	90.3072	88.4848	1.8224	−0.030672
B20	82.3034	82.3033	0.0001	0.000098
B40	82.5386	82.4958	0.0428	0.005378
B60	84.2135	83.326	0.8875	0.776491

**Table 7 materials-13-05159-t007:** Parameter analysis of the quadratic curve fitting.

Parameter	A	R-Squared
The experiment’s curve of Group A	4.58883 × 10^−4^	0.89917
The FEA’s curve of Group A	5.38266 × 10^−4^	0.96372
The experiment’s curve of Group B	2.08133 × 10^−4^	0.77993
The FEA’s curve of Group B	2.93054 × 10^−5^	0.95966

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
