# Peer review of "Thermal Ablation Damage Analysis of CFRP Suffering from Lightning Based on Principles of Tomography"

_materials, 2020, doi:10.3390/ma13225159_

Round 1

Reviewer 1 Report

The manuscript “Thermal Ablation Damage Analysis of CFRP Suffering from Lightning Based on Principles of Tomography” proposes the use of the computed tomography technique to analyze the thermal ablation damage of the carbon fiber reinforced polymer (CFRP) caused by lightning.

According to the authos, there are advantages to using this approach, such as the possibility of analyzing complex spatial geometries. -The work developed is clearly important from an academic and industrial point of view. It was well done, in general, and adds intersting conclusions. I have some sugestions: -The last paragraph of the introduction topic should always make it clear what the purpose of the work is. The main objective... - In line 278 presents the mensage Error! Reference source not found.. -The mass losses and the evolution of the morphologies were analyzed by the computed tomography technique. It is a very interesting approach to study the damage caused by the thermal ablation of the carbon fiber reinforced polymer caused by lightning. Alternatively there is another method which could be used to investigate this issue, 3-D microstructure reconstruction, based on digital microscopy*.CT has the advantage of being a non-destructive method, on the other hand, 3-D microstructure reconstruction would provide a more realistic analysis. I recommend that you add this alternative form, in the topic of the introduction, which is well explained in the cited article. * D.Mata, A.L. Horovistiz, I. Branco, M.Ferro, N.M. Ferreira, M.Belmonte, M.A. Lopes, R.F. Silva, F.J. Oliveira, Carbon nanotube-based bioceramic grafts for electrotherapy of bone, Materials Science and Engineering: C, v 34, 2014, pp 360-368

Author Response

Dear reviewer,

I have revised the paper according to your opinion.

1) I have made the main purpose of this article clearer on lines 74-76.

2) I have delete the message "Error! Reference source not found." and changed it to "[38]".

3) I have read the paper you supplied and introduced the research methods inside on lines 69-70.

Thanks for your valuable advice!

Reviewer 2 Report

Dear Author,

The manuscript under the title: “Thermal Ablation Damage Analysis of CFRP Suffering from Lightning Based on Principles of Tomography”  is relevant for the Materials journal. The authors work on up-to-date topic connected with carbon fiber reinforced polymer materials. The article based on original experimental research. The organization of the article is appropriate. The abstract is sufficiently informative. Overall, the paper is well prepared. The article required only some small improvements:

Introduction: Please provide some information regarding the impurities that can appear in carbon fiber reinforced polymers

I think some extra references from the field of carbon fiber reinforced polymer can be added. 

Overall it is a well-prepared paper and with minor corrections can be published.

Author Response

Dear reviewer,

I have revised the paper according to your opinion.

1) I have explained the possible impurities in CFRP, and analyzed its content and its impact on lines 43-45.

2) I added a brief introduction to CFRP lightning protection methods, and pointed out that copper mesh protection is the main protection method on lines 62-63.

Thank you for your valuable comments!